# Fabrication of AuNPs/MWCNTS/Chitosan Nanocomposite for the Electrochemical Aptasensing of Cadmium in Water

**DOI:** 10.3390/s22010105

**Published:** 2021-12-24

**Authors:** Selma Rabai, Ahlem Teniou, Gaëlle Catanante, Messaoud Benounis, Jean-Louis Marty, Amina Rhouati

**Affiliations:** 1Laboratory of Sensors, Instrumentations and Process (LCIP), University of Khenchela, Khenchela 40000, Algeria; gp.selma@yahoo.fr (S.R.); benounis@yahoo.fr (M.B.); 2Bioengineering Laboratory, Higher National School of Biotechnology, Constantine 25100, Algeria; teniouahlem97@gmail.com (A.T.); jlmarty@univ-perp.fr (J.-L.M.); 3Laboratoire Biocapteurs, Analyse et Environnement (BAE), Université de Perpignan Via Domitia, 66860 Perpignan, France; gaelle.catanante@univ-perp.fr

**Keywords:** cadmium, aptasensor, electrochemical, environment, pollutant

## Abstract

Cadmium (Cd^2+^) is one of the most toxic heavy metals causing serious health problems; thus, designing accurate analytical methods for monitoring such pollutants is highly urgent. Herein, we report a label-free electrochemical aptasensor for cadmium detection in water. For this, a nanocomposite combining the advantages of gold nanoparticles (AuNPs), carbon nanotubes (CNTs) and chitosan (Cs) was constructed and used as immobilization support for the cadmium aptamer. First, the surface of a glassy carbon electrode (GCE) was modified with CNTs-CS. Then, AuNPs were deposited on CNTs-CS/GCE using chrono-amperometry. Finally, the immobilization of the amino-modified Cd-aptamer was achieved via glutaraldehyde cross-linking. The different synthesis steps of the AuNPs/CNTs/CS nano assembly were characterized by cyclic voltammetry (CV). Electrochemical impedance spectroscopy (EIS) was employed for cadmium determination. The proposed biosensor exhibited excellent performances for cadmium detection at a low applied potential (−0.5 V) with a high sensitivity (1.2 KΩ·M^−1^), a detection limit of 0.02 pM and a wide linear range (10^−13^–10^−4^ M). Moreover, the aptasensor showed a good selectivity against the interfering ions: Pb^2+^; Hg^2+^ and Zn^2+^. Our electrochemical biosensor provides a simple and sensitive approach for Cd^2+^ detection in aqueous solutions, with promising applications in the monitoring of trace amounts of heavy metals in real samples.

## 1. Introduction

Aptamers are short oligonucleotidic sequences identified by an in vitro process called SELEX, the “Systematic Evolution of ligands by Exponential enrichment” [1]. They are generated by a simple chemical synthesis with extreme precision and reproducibility [2]. Aptamers are characterized by a high specificity, selectivity and stability, making them strong competitors of antibodies in molecular analysis [3,4].They thus constitute good candidates as effective probes for biosensing [5,6,7]. One of the key challenges in the application of aptamers in analysis is to ensure an efficient and cost-effective immobilization method. In order to immobilize aptamers in a functional manner, the conjugation process must not interfere with aptamer folding [8]. As they are chemically synthesized, aptamers can be easily labeled to incorporate functional groups in 3′ or 5′ end. Different modifications can be used, such as thiol groups, to allow immobilization on gold electrodes and carboxylic or amine groups for carbon electrodes. Among the different strategies, the immobilization of aptamers onto nanomaterials and nanostructured materials has gained much attention in recent years.

Nanotechnology is playing an increasingly important role in different fields, including sensing and biosensing [9,10]. The use of nanomaterials is advantageous; whether for sensors miniaturization, signal amplification or for developing new immobilization strategies for biomolecules [11,12]. Among the wide range of nanomaterials, carbon and metallic nanostructures have attracted particular attention. Carbon nanomaterials, such as carbon nanotubes (CNTs) and nanofibers, graphene and nano diamond, are characterized with excellent analytical and electroanalytical properties. CNTs are particularly interesting in electrochemical sensing because of their remarkable electrical conductivity. Their length/diameter aspect ratio provides a high surface/volume ratio. Indeed, CNTs have a unique ability to promote fast electron transfer kinetics for a wide range of electroactive species [13,14]. CNTs-modified electrodes have been widely used as promising anchors for bioreceptors (enzymes, proteins, nucleic acids, etc.) because of their biocompatibility, sensitivity and improved signal to noise ratio [15]. In parallel, carbon nanotubes can be easily functionalized by other nanomaterials, chemical groups or covered with different polymers. In recent years, CNTs decorated with metallic nanoparticles demonstrated a great potential in the development of sensing platforms. Owing to their unique properties, gold nanoparticles (AuNPs) have been widely integrated in CNTs-based nanohybrids [16,17,18]. AuNPs are characterized with a high electrochemical potential range, improving electron transfer and contributing tothe increased sensitivity of biosensors. They provide a minimized diffusion problem with an increased contact surface and graft density of biomolecules. In addition, they maintain the stability and biological activity of biomolecules [19,20]. The synergistic contributions of carbon nanotubes and gold nanoparticles have been widely explored in chemical sensing strategies [17,21]. Some reports explored AuNPs-CNTs nanohybrids as immobilization support for enzymatic bioreceptors [22,23]. However, just one report describes the combined application of AuNPs and CNTs in aptasensing [24].

In this work, we aim to investigate the advantages of CNTs and AuNPs in the immobilization of an amino-modified aptamer. The adopted procedure is based on the modification of a glassy carbon electrode (GCE) with dispersed carbon nanotubes in Chitosan (CS). Once grafted around carbon nanotubes, chitosan improves their solubility and biocompatibility [25]. Then, gold nanoparticles were electro grafted on CNTs-CS/GCE by using chrono-amperometry. The synergistic effect of CNTs and Au-NPs will make the electron transfer easier with a good conductivity and high electroactive surface area, thus providing a biocompatible microenvironment for the aptamer. Afterwards, the biorecognition element was immobilized by cross-linking the amino-modified aptamer and CS through glutaraldehyde onto the AuNPs/CNTs/CS modified surface [25,26]. The different steps of the aptasensing platform fabrication were characterized electrochemically by cyclic voltammetry (CV).

In order to evaluate the analytical performance of the functionalized surface, cadmium ions (Cd^2+^) were selected as a target analyte. Cadmium is a toxic heavy metal affecting lungs, kidneys, bones and the immune system. The toxicity of cadmium occurs by an oral or respiratory route [27]. It can lead to lung and prostate cancer, cause cardiovascular diseases and anemia, as it seems to contribute to autoimmune thyroid diseases. Because of this, the World Health Organization has designated cadmium as a human carcinogen which can lead to death [28]. In the literature, few reports describe the electrochemical aptasensing of this hazardous heavy metal. In a previous study, we developed a biosensing platform, where diazonium chemistry was used for immobilizing cadmium aptamer [29]. In this work, a new immobilization strategy was adopted by exploring the unique properties of AuNPs and carbon nanotubes. Electrochemical impedance spectroscopy (EIS) was used to measure the extent of binding between cadmium and its specific aptamer immobilized on the AuNPs/CNTs/CS/GCE surface. First, the constructed biosensing platform was used to detect cadmium in phosphate-buffered saline, where excellent analytical performances have been obtained. We reached the low detection limit of 0.02 pM and a wide linear range, varying from 10^−3^ M to 10^−13^ M. The biosensor selectivity for cadmium detection was also investigated against a number of competing metal ions, including lead, mercury and zinc. Finally, the applicability of the label-free aptasensor has been demonstrated in real water samples.

## 2. Experimental

### 2.1. Chemical and Reagents

Cadmium (II) nitrate (Cd (NO_3_)_2_), tetra chloroauric acid (HAuCl_4_), glutaraldehyde (25%), chitosan (deacetylation ≥ 75%), phosphate buffered saline (PBS), potassium ferrocyanide (II), potassium ferricyanide (III), and sodium sulfate (Na_2_SO_4_), were purchased from Sigma–Aldrich, France. Carbon nanotubes (CNTs) were obtained from Sigma-Aldrich, France, and were functionalized with carboxyl groups (COOH). Cadmium aptamer was synthesized and purchased from Microsynth, Switzerland. The specific sequence of the aptamer was as follows:

Aptamer sequence (5′amino-modified): 5′-ACC GAC CGT GCT GGA CTC TGG ACT GTT GTG GTA TTA TTT TTG GTT GTG CAG TAT GAG CGA GCG TTG CG-3′ [30].

### 2.2. Instrumentation

The electrochemical experiments were carried out using a potentiostat (Biologic EC-Lab SP-300, Seyssinet-Pariset, France). All experiments were performed at room temperature. The modeling of the obtained EIS data was achieved by the EC-Lab software using the Randomize + Simplex method. Herein, randomizing was stopped on 100,000 iterations and the fit stopped on 5000 iterations. Electrochemical experiments were carried out using a conventional three-electrodes system with a GCE (U = 3 mm) as the working electrode, a platinum wire as the auxiliary electrode, and an Ag/AgCl/3.0 M KCl as the reference electrode. Aptamer heating was performed using a thermocycler from Eppendorf.

### 2.3. Fabrication of the Aptasensor

Figure 1 illustrates the different steps of fabrication of the aptamer/CNTs/AuNPs modified GCE surface. First, GCE was modified by a drop of 6 µL of 0.5 mg mL^−1^ CNTs–CS (1 mg of CNTs dispersed in 2.0 mL of 0.2% CS solution) and kept to dry in air. Then, AuNPs were electrodeposited on a CNTs–CS modified electrode with potentiostatic electro-deposition of 0.2 M Na_2_SO_4_ solution containing 1 mM of HAuCl_4_ for 400 s at −0.2 V. After the electrodeposition of NPs, the modified electrode was gently rinsed in water and air-dried at room temperature. For aptamer immobilization, 6 µL of CS solution (0.2%) were added to the AuNPs/CNTs/CS film and air-dried at room temperature. Then the modified electrode was immersed in a 0.25% glutaraldehyde solution for 2 h followed by the surface incubation with 5 µL of aptamer solution overnight at 4 °C. Some pictures of the real sensor and steps of fabrications are provided in the Appendix A.

### 2.4. FTIR Characterization

To further confirm the immobilization of AuNPs on the MWCNT/Chitosan, a FT-IR spectrometer (Shimadzu, Noisiel, France) equipped with a crystal ATR single reflection diamond-sampling module was used. Infrared spectra were collected at an average of five scans per sample between the wave number range of 4000–500 cm^−1^ at a resolution of 4 cm^−1^, and LabSolutions IR Software was used to analyze spectra. The analysis was kept simple, since at first the spectrum of a bare substrate was recorded and considered as a reference. In the second step, the spectrum of substrates having MWCNTs and AuNPs-decorated MWCNT films were recorded and, finally, the spectrum of an aptamer-functionalized, AuNPs/MWCNT film was recorded.

### 2.5. Analytical Performance of the Label-Free Aptasensor

Once the aptamer immobilized on the electrode surface, the resulting aptasensor was tested for cadmium detection. For this, the modified surface was incubated with increasing concentrations of cadmium prepared in PBS buffer for 30 min. Then, the electrochemical response was monitored by EIS. In parallel, the aptasensor selectivity was studied by incubating the sensing surface with different concentrations of other interfering heavy metals, Hg^2+^, Pb^2+^ and Zn^2+^, over 30 min.

### 2.6. Real Sample Analysis

In order to validate and evaluate the accuracy and applicability of our aptasensor, analyses were carried out on real river water polluted by public work wastes and containing nitrate and cadmium [31]. This river is crossed by a volcanic water source from the Hammam Essalihine valley (Khenchela, Algeria) which is rich in minerals, such as bicarbonates, sulphates, calcium and magnesium. We have obtained samples from the river containing cadmium and other potential interferences.

## 3. Results and Discussion

### 3.1. FT-IR Characterization of the Fabrication Steps

FTIR analysis were performed to study the different steps of the aptasensor fabrication. Figure 2 shows the FTIR spectra of the activated CNTs (curve a), activated CNTs-AuNPs (curve b), and activated CNTs-AuNPs-Aptamer (curve c). As shown in Figure 2, The FTIR spectrum of activated CNTs (curve a) presents a characteristic carboxyl peak at 1716.64 cm^−1^ which can be attributed to C=O stretching. Therefore, we can conclude that the CNTs have been successfully activated by introducing the carboxylic group. Moreover, the vibration peak at 1558.48 cm^−1^ confirms the presence of C-C double bond that forms the framework of the carbon nanotube sidewall [32,33]. A strong peak was also observed at 1066.63 cm^−1^; this is probably associated to the stretching of sulfoxide bond (S-O), usually created in defective CNT areas. As was reported by Chiang’s group, a high H_2_SO_4_ concentration may lead to the remaining of sulfoxide groups between two free carbons at the end, defect carbons or two different MWCNTs [34]. On the other hand, the strong (large) band observed between 2900–3500 cm^−1^ can be attributed to the O-H stretching mode of the carboxyl groups. However, the sharp peak at 1406 cm^−1^ may be related to the nitro group formed during high-pressure HNO_3_ treatment under heating conditions [35]. After the electrodeposition of AuNPs, FTIR spectra was also recorded in curve b in Figure 2. As compared to the spectra shown in curve (a), the strong band between 2900–3500 cm^−1^ related to the O–H group was weakened and shifted (2800–2250 cm^−1^). This can be related to the attachment of AuNPs to the surface of CNTs [36]. Based on thesmall shoulder observed in the range of 500 to 750 cm^−1^, the covalent attachment of AuNPs could be confirmed, and thiscan be assigned to stretch the mode of C−S groups [37]. Finally, the last step of the aptasensor fabrication corresponding to the aptamer immobilization onto the MWCNTs-AuNPs composite was also explored by FTIR analysis. The obtained spectra in curve c in Figure 2 shows additional bands at 1558 cm^−1^, confirming the aptamer binding.

### 3.2. Electrochemical Characterization of the Aptamer-Modified Surface

Cyclic voltammetric characterization of electroactive species is a valuable tool for testing the kinetic barrier and electrochemical exchange at the interface [38]. In this work, the different steps of the surface modification have been characterized using cyclic voltammetry where all experiments have been performed in the presence of the redox couples K_3_[Fe(CN)_6_]/K_4_[Fe(CN)_6_] by scanning the potential between −1 V and 1 V, with a scan rate of 50 mV s^−1^. Figure 3 represents the electrochemical characterization of the bare and modified glassy carbon electrode by cyclic voltammetry. First, it can be seen from the voltammograms that the oxidation and reduction peaks of the redox couple are visible in the bare electrode, shown in curve a in Figure 3, due to the high exchange rate with the glassy carbon electrode. Then, after the modification of the electrode with CNTs/CS, shown in curve b in Figure 3), a voltametric effect of ferricyanide on the electrode was observed. Thiscan be explained by the active catalytic surface of carbon nanotubes which may increase the active surface of the modified electrode [39]. However, the oxidation and reduction peaks of the redox couple decreased, as compared to that observed with the bare GCE, due to the blocking behavior of CS [40]. After the electrodeposition of AuNPs (Curve c in Figure 3) on the CNTs-CS/GCE surface, we noted that the oxydo/reduction peaks increased and the ΔEp decreased as compared to the CNTs/CS modified electrode. These results confirm the role of AuNPs in increasing the electroactive catalytic surface. We can see clearly that the highest peak was obtained with the AuNPs/CNT-CS composite, because of the large specific surface of the modified electrode and the synergistic effect of AuNPs and CNTs [41,42]. Finally, we can see from curve d in Figure 3, corresponding to the immobilization of the aptamer on the nanocomposite-modified GCE that the oxydo/reduction peaks, as well as the ΔEp, have significantly decreased as compared to the AuNPs/CNTs-CS/GCE. This implies that most of the electrode surface was successfully covered with aptamer [43]. This decrease returns mainly to the negative charge of the phosphate backbone of the DNA aptamer acting as a barrier hindering the electron transfer. After incubating the aptasensing platform with a buffer solution (pH = 4.5) containing 0.1 mM cadmium Cd^2+^ during 30 mn, CV showed a further peak decrease and an increase in ΔEp, in curve e in Figure 3. This behavior was attributed to the adsorption of Cd^2+^ on the surface, which considerably hampers the transfer of electrons between the surface of the electrode and the redox couples in solution due to the opening of the rod portion of aptamer and formation of the complex Cd^2+^-aptamer [44,45].

### 3.3. Application of the Aptasensing Platform in the Impedimetric Detection of Cadmium

The ability of the proposed aptasensor for the quantification of cadmium was evaluated. For that, the aptamer-modified surface was exposed to increasing concentrations of cadmium, under the optimal conditions. The experimental results showed a significant increase in the diameter of the Nyquist circle up on cadmium addition on the modified electrode surface. As can be seen from Figure 4A, the charge transfer resistance (Rct) values enhanced by increasing Cd^2+^ concentration in a logarithmic way. The first semi-circle of the Nyquist plot corresponds to the aptamer-modified surface before the addition of cadmium. After incubation with increasing concentrations of the target, the Nyquist plot semi-circles continue to increase, indicating the interaction between cadmium and its specific aptamer.

Nyquist plot semi circles were fitted using Randles equivalent circuit model (Figure 4B) where Rs represents the solution resistance, Rct represents the charge-transfer resistance and Q1 represents the constant phase element (an equivalent model of double-layer capacitance). The real part of impedance (Re(Z)) increased as function of cadmium concentration. The results demonstrated that the more Cd^2+^ was bound on the aptasensing platform, the higher the Rct was. The linear regression equation was normalized to ∆Rct = Rct (aptamer-cadmium) –Rct (aptamer) as a function logarithm of cadmium concentration. Rct (aptamer-cadmium) is the value of the electron transfer resistanceafter Cd^2+^ binding to the immobilized aptamer on the modified electrode surface. Rct (aptamer) represents the value of the electron transfer resistance after the aptamer immobilization on the modified electrode surface. A steady linear relationship between the Rct and cadmium concentrations was found in the range of concentrations (10^−13^–10^−4^ M), with a good correlation coefficient of (R^2^ = 0.985) and a sensitivity of 1.2 KΩ·M^−1^ per cadmium concentration decade. The limit of detection (LOD) of the aptasensor was 2.011 × 10^−14^ M, calculated as the concentration of cadmium corresponding to the three times “s/m” ratio, where “s” is the standard deviation of the blank impedance signal (three replicates) and “m” is the slope of the related calibration curve [46].

### 3.4. Specificity and Reproducibility

The specificity of the proposed aptasensor was evaluated by comparing the impedance changes after incubation with Cd^2+^ and other heavy metals, such as Hg^2+^, Pb^2+^ and Zn^2+^, during 30 min. Three concentrations have been tested for each analyte; 10^−8^, 10^−7^ and 10^−6^ M. At this stage, the interactivity of the biosensor and cadmium detection was performed only in PBS buffer. As can be observed in Figure 5, the impedance measurements show clearly that the biosensor was highly selective to cadmium as compared to the other tested heavy metals. No significant signals were recorded in the presence of Pb^2+^ and Zn^2+^. However, a slight cross-reactivity was noted with Hg^2+^ which may be attributed to the special T–Hg–T mismatched base pair between Hg^2+^ and Cd-aptamer [47,48,49]. These results show that the aptamer immobilized on the electrode surface maintains its affinity and specificity in reaction with the cadmium, producing a significant ∆Rct increase. This may be considered as a validation of the selectivity of the proposed aptasensor, because it allows the discrimination between the tested heavy metals into the entire selected range.

The reproducibility of the developed aptasensing platform was tested with inter assay precision. The inter assay precision was evaluated by incubating three electrodes, independently prepared in the same experimental conditions with the same concentration of cadmium. The coefficient of variation obtained from three measurements was good, varying between 2% and 4% in the range of concentrations studied.

### 3.5. Applicability in Real Water Samples

The selective cadmium ion biosensor was used to test a water sample taken from Khenchela region, which represents an area of water contaminated with cadmium. The water sample was obtained from the Hammam Essalihine (Khenchela, east of Algeria). The initial concentration of cadmium ions was determined by atomic absorption spectroscopy (data not shown). Then, buffer dilutions (pH 4.5) were prepared with these water samples. Afterwards, the sensor was subjected to different concentrations of cadmium prepared in water samples, and its response was revealed using the electrochemical impedance spectroscopy method. Figure 6A showsthe Nyquist plots corresponding to the aptamer-modified surface without cadmium addition, and after incubation with different concentrations of cadmium. A shift in the semi-circle with the first concentration (10^−8^ M) was observed, and attributed to the increase of resistance provided from an aptamer reaction with cadmium. The analysis of the other concentrations was performed in the same parameters. An increase in the semi-circular diameter was noted by increasing the cadmium concentration. The signal was improved and the analytical response of the biosensor was linear, within the range of 10^−8^ to 10^−12^ M with a good correlation coefficient of (R^2^ = 0.98) and a LOD of 10^−12^ M (Figure 6B). These results confirm the feasibility of our label-free aptasensing strategy in real samples.

## 4. Conclusions

In summary, a label-free aptasensor was successfully developed for cadmium determination in water samples. A new immobilization strategy was adopted in the construction of the sensing platform. For that, the aptamer was crosslinked on a glassy carbon electrode via a nanocomposite of carbon nanotubes, gold nanoparticles and chitosan. Due to the synergistic effect of CNTs and AuNPs, the as synthesized nanohybrid offered a biocompatible support for the DNA aptamer. In addition, it exhibited an excellent conductivity and electrocatalytic activity, thus enhancing the analytical performance. The label-free aptasensor has enabled the sensitive detection of cadmium with the detection limit of 0.02 pM. The obtained response was linear, in the range of [10^−13^–10^−4^ M]. Moreover, the aptasensor has shown a high selectivity to cadmium in the presence of other interfering metals, including Hg^2+^, Pb^2+^ and Zn^2+^. Finally, the constructed aptamer biosensor has been applied to detect cadmium in real samples with high accuracy. On the whole, this work opens a new avenue for aptamer immobilization on CNTs/AuNPs nanocomposites and their application in electrochemical biosensing.

## Figures and Tables

**Figure 1 sensors-22-00105-f001:**
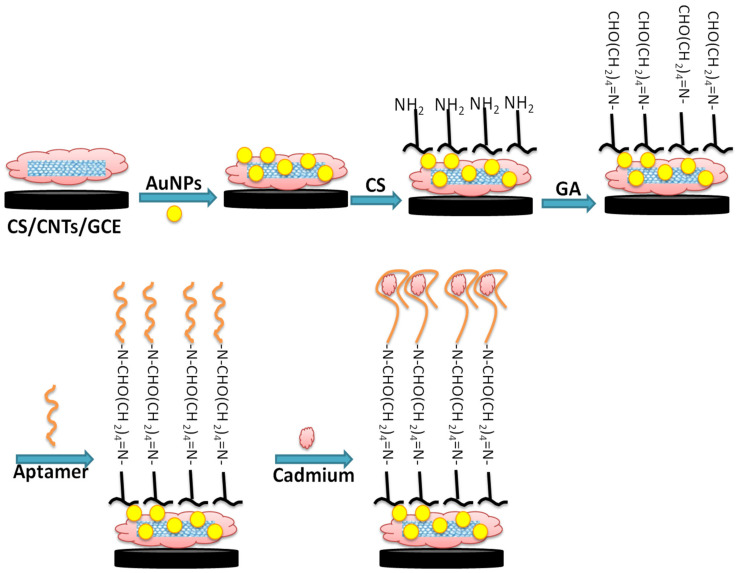
Schematic illustration of GCE chemical surface modification with CNTs-AuNPs/CS nanocomposite for aptamer immobilization and application in cadmium detection.

**Figure 2 sensors-22-00105-f002:**
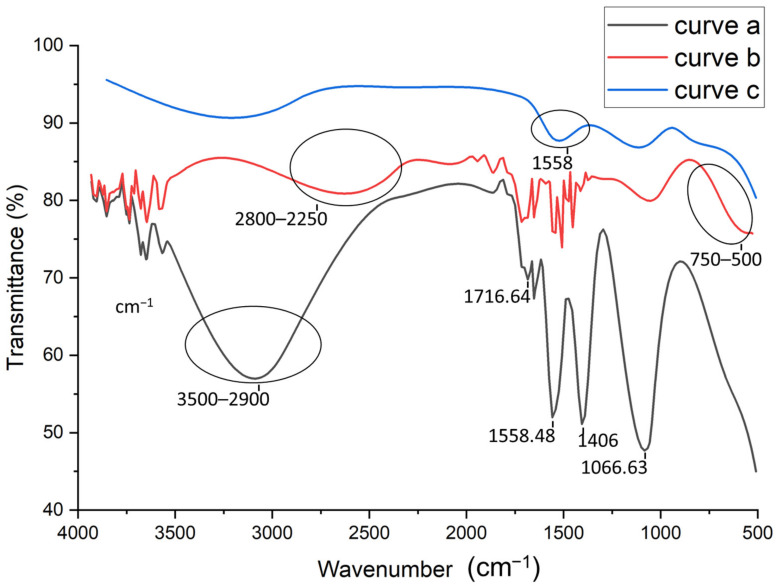
FTIR spectra for: activated carbon nanotubes (CNTs) (curve a), Gold nanoparticles (AuNPs)/CNTs (curve b), Aptamer/AuNPs/CNTS (curve c).

**Figure 3 sensors-22-00105-f003:**
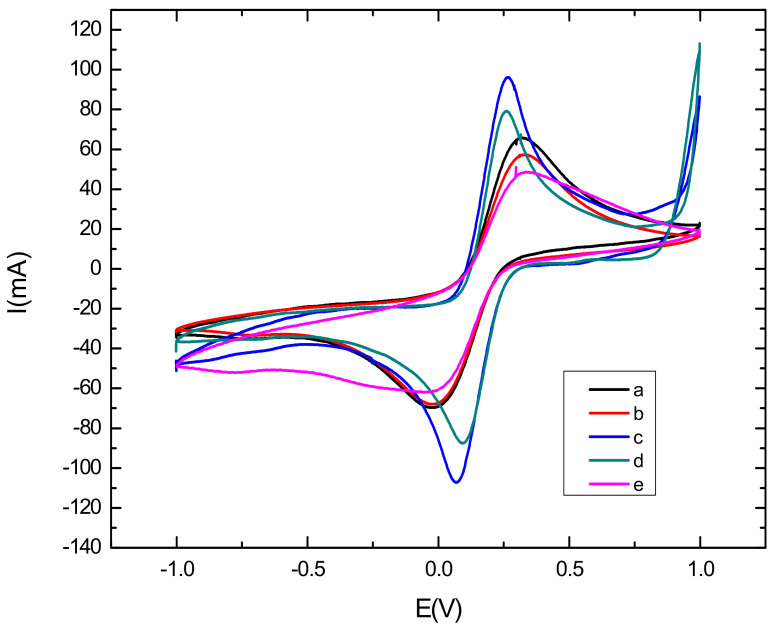
Cyclic voltammograms in the presence of K_3_[Fe(CN)_6_]/K_4_[Fe(CN)_6_] for: bare electrode (curve a), Cs-CNTs/GCE (curve b), AuNPs/Cs-CNTs/GCE (curve c), Apta/AuNPs/Cs-CNTS/GCE (curve d), Cd^2+^/Apta/AuNPs/Cs-CNTS/GCE (curve e). Potential was scanned from 1 V to −1 V with a scan rate of 50 mV s^−1^.

**Figure 4 sensors-22-00105-f004:**
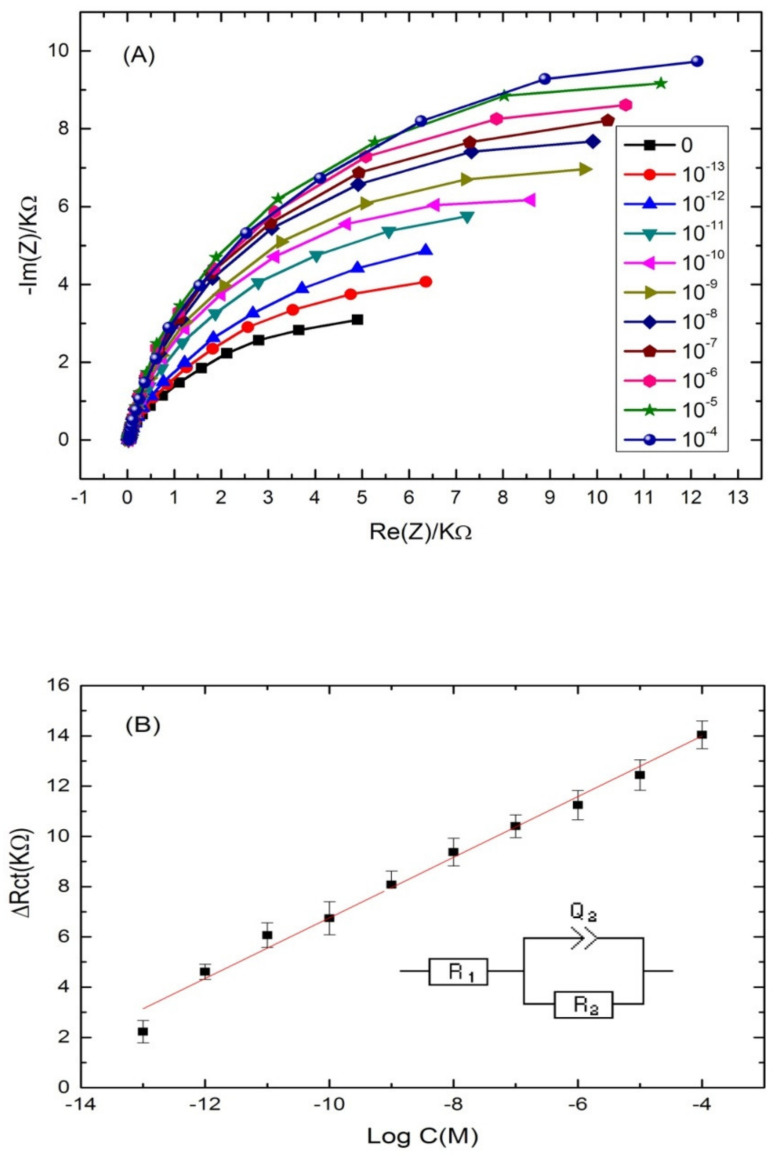
(**A**) Nyquist plots of Cd-aptamer-modified surface incubated with increasing concentrations of cadmium ranging from 10^−4^ to 10^−13^ M. Measurements were performed in buffer (pH 4.5) at a potential of −0.5 V and a frequency range from 10 kHz to 100 mHz. (**B**) Calibration plots of the increase of the charge transfer resistance with cadmium concentration for the label free aptasensor. Inset: equivalent circuit used for the EIS equivalent circuit fitting.

**Figure 5 sensors-22-00105-f005:**
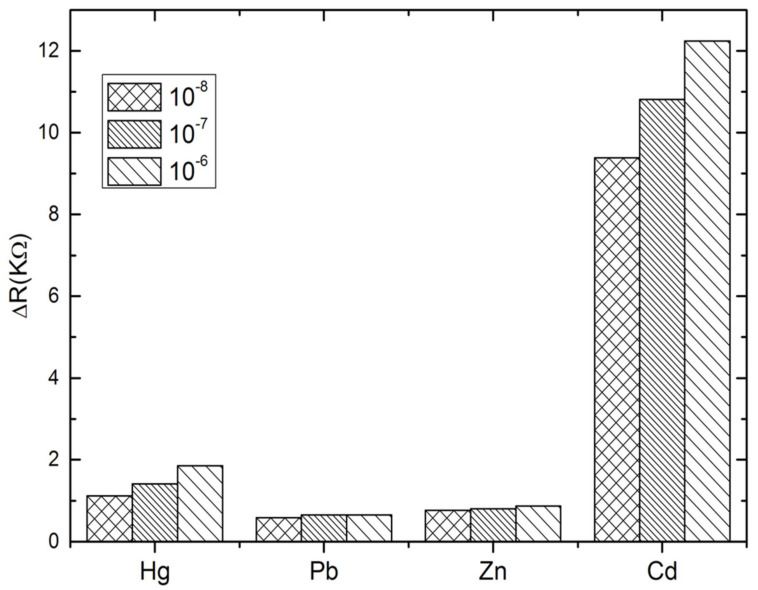
Selectivity studies of the aptasensor with mercury (Hg^2+^), lead (Pb^2+^) and Zinc (Zn^2+^).

**Figure 6 sensors-22-00105-f006:**
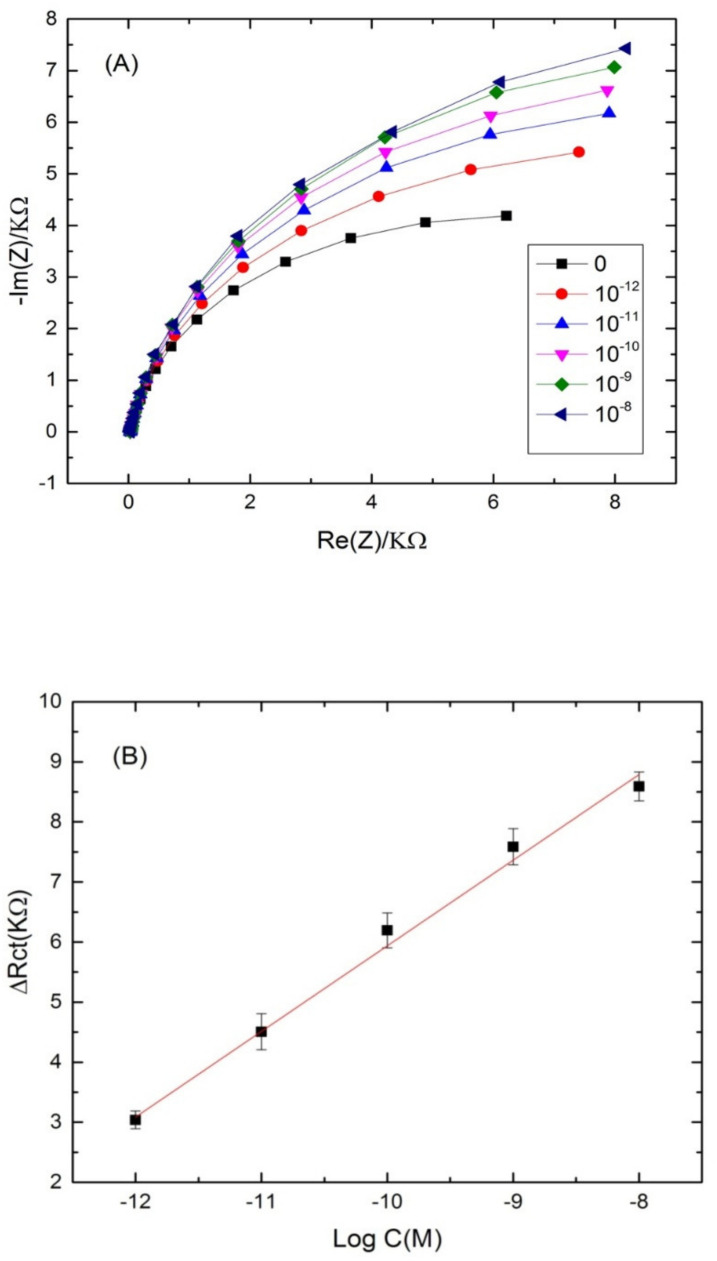
(**A**) Nyquist plots of Cd-aptamer-modified surface incubated with increasing concentrations of real samples spiked with cadmium. Measurements were performed at a potential of −0.5 V and a frequency range from 10 kHz to 100 mHz. (**B**) Calibration plots of the increase of the charge transfer resistance with cadmium concentration in real water samples for the label free aptasensor.

## Data Availability

Not applicable.

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
