# Peer review of "Fabrication of AuNPs/MWCNTS/Chitosan Nanocomposite for the Electrochemical Aptasensing of Cadmium in Water"

_sensors, 2021, doi:10.3390/s22010105_

Round 1

Reviewer 1 Report

see attached file

Author Response

  1. SEM/EDX or TEM analyses could give further information on the morphology and the effective chemical composition of the electrodes at each modification step.

As SEM and TEM are not available in our institute, FTIR characterization was performed to study the different fabrication steps.

  1. The measurements with different electrodes should be reported to study the reliability of the proposed method.

It is mentioned in results and discussion section that reliability was assessed with different electrodes:

“The interassay precision was evaluated by incubating three electrodes, independently prepared in the same experimental conditions, with the same concentration of cadmium. The coefficient of variation obtained from three measurements was good varying between 2% and 4% in the range of concentrations studied”.

  1. In the section dedicated to real water samples the authors refer also to cadmium determination performed through atomic absorption spectroscopy; a comparison between the spectroscopic and the proposed electrochemical method should be reported to check its reliability.

Before applying the developed method in real water samples, the collected samples were analyzed by atomic absorption spectroscopy (AAS) to confirm that the water is free from cadmium ions. AAS was selected because it is considered as a reference technique for heavy metals analysis. The object of this analysis was not to compare the two techniques. For that, the results were not represented and discussed  in the manuscript.

  1. 2 control of the caption is needed.

Caption was corrected

Reviewer 2 Report

The manuscript describes about "Fabrication of AuNPs/MWCNTS/Chitosan nanocomposite for the electrochemical aptasensing of cadmium in water". However, the authors did not provide any evidence of the fabrication of the composite (photograph, SEM micrograph, XRD analysis...). Only the graphs and scheme cannot persuade the readers about the soundness of this research. Furthermore, there are many flaws in the figures, for example: the quality of Fig.1 is too low to read the functional groups; legend in Fig. 5(A) was crossed over by an arrow... 

My recommendation is the authors should change the title if they want to contribute to the issue, because the content of the manuscript does not show any evidence of the "fabrication of the composite". Otherwise, they must provide evidence of the fabrication (SEM, TEM, XRD, FT-IR analysis...) 

Author Response

The manuscript describes about "Fabrication of AuNPs/MWCNTS/Chitosan nanocomposite for the electrochemical aptasensing of cadmium in water". However, the authors did not provide any evidence of the fabrication of the composite (photograph, SEM micrograph, XRD analysis...). Only the graphs and scheme cannot persuade the readers about the soundness of this research. Furthermore, there are many flaws in the figures, for example: the quality of Fig.1 is too low to read the functional groups; legend in Fig. 5(A) was crossed over by an arrow... 

My recommendation is the authors should change the title if they want to contribute to the issue, because the content of the manuscript does not show any evidence of the "fabrication of the composite". Otherwise, they must provide evidence of the fabrication (SEM, TEM, XRD, FT-IR analysis...).

  1. Figure 5 was improved.
  2. As SEM, DRX and TEM are not available in our institute; FTIR characterization was performed to study the different fabrication steps.

Round 2

Reviewer 1 Report

see attached file

Author Response

Reviewer 1:

The authors have significantly improved the manuscript from the previous version. The detailed statement and corrections make the work acceptable to be published in Sensors. This paper is useful for most researchers working on modified electrodes for electrochemical apta-sensing of ions in solutions. I recommend it for publication in the present form.

Thank you very much

Reviewer 2 Report

The manuscript was revised and a new figure (Fig. 2) was added. However, only this FT-IR result is not important enough to prove that the nanocomposite has been successfully fabricated. In addition, there are still many typos (for example: "cm-1") and low definition figures (Fig.1, Fig. 4, etc.).

Author Response

Reviewer 2:

The manuscript was revised and a new figure (Fig. 2) was added. However, only this FT-IR result is not important enough to prove that the nanocomposite has been successfully fabricated. In addition, there are still many typos (for example: "cm-1") and low definition figures (Fig.1, Fig. 4, etc.).

  • The objective of our work is the fabrication of MWCNTS/AuNPs/Chitosan on the electrode surface for the immobilization of an amino-modified aptamer. FT-IR characterizations have shown the covalent attachment of AuNPs to the surface of CNTs. FTIR has also confirmed the aptamer immobilization onto the MWCNTs-AuNPs composite. In parallel, all the functionalization steps of the electrode surface were characterized electrochemically. Cyclic voltammetric characterization, in ferro/ferricyanide, is considered as a valuable tool for testing the kinetic barrier and confirming the electrodeposition of the different layers on the electrode surface. After the addition of AuNPs on the CNTs-CS/GCE surface, we noted a remarkable increase in oxydo/reduction peaks confirming the successful electrodeposition of AuNPs on the electrode surface which allowed an efficient immobilization of the aptamer in the following step. For that, we estimated that FT-IR and electrochemical characterization was sufficient to confirm the deposition of our composite on the glassy carbon electrode.
  • Typos were corrected
  • Figures quality was improved

Round 3

Reviewer 2 Report

The manuscript has been revised again and its quality is better than previous version. I suggest the authors provide some pictures of the fabrication of the sensor (electrodeposition, surface incubation...), and pictures of the real sensor.

Check the space between words (for example, page 6: "fromFig.3d"...)

There are two Fig.4.

The legend in Fig.6(a) should be presented in a similar way as in Fig.4(A), not by an arrow.

Author Response

1/ I suggest the authors provide some pictures of the fabrication of the sensor (electrodeposition, surface incubation...), and pictures of the real sensor.

-Pictures have been added as supporting material

2/Check the space between words (for example, page 6: "fromFig.3d"...)

Spaces have been checked

3/There are two Fig.4.

Mistake was corrected

4/The legend in Fig.6(a) should be presented in a similar way as in Fig.4(A), not by an arrow.

The required modification was done